# Observation of spin-current striction in a magnet

Hiroki Arisawa [1✉], Hang Shim[2], Shunsuke Daimon [3,4], Takashi Kikkawa [1,3,5], Yasuyuki Oikawa[5], Saburo Takahashi[5], Takahito Ono[2,6] & Eiji Saitoh[1,3,4,5]

The interplay among magnetization and deformation of solids has long been an important issue in magnetism, the elucidation of which has made great progress in material physics. Controlling volume and shapes of matter is now indispensable to realizing various actuators for precision machinery and nanotechnology. Here, we show that the volume of a solid can be manipulated by injecting a spin current: a spin current volume effect (SVE). By using a magnet $Tb_{0.3}Dy_{0.7}Fe_2$ exhibiting strong spin-lattice coupling, we demonstrate that the sample volume changes in response to a spin current injected by spin Hall effects. Theoretical calculation reflecting spin-current induced modulation of magnetization fluctuation well reproduces the experimental results. The SVE expands the scope of spintronics into making mechanical drivers.

[1] Institute for Materials Research, Tohoku University, Sendai 980-8577, Japan. [2] Department of Mechanical Systems Engineering, Tohoku University, Sendai 980-8579, Japan. [3] Department of Applied Physics, The University of Tokyo, Tokyo 113-8656, Japan. [4] Institute for AI and Beyond, The University of Tokyo, Tokyo 113-8656, Japan. [5] WPI, Advanced Institute for Materials Research, Tohoku University, Sendai 980-8577, Japan. [6] Micro System Integration Center (μ-SiC), Tohoku University, Sendai 980-8579, Japan. ✉email: h.arisawa@imr.tohoku.ac.jp

The magneto-volume effect[1-3] (MVE), one of the magnetostriction effects, has been a central issue in the physics of magnetism in itinerant electron systems for a long time. One notable example is an invar alloy $Fe_{64}Ni_{36}$[4-6], where the magneto-volume change compensates for thermal expansion; the volume change is attributed to spin fluctuation in the alloy (see Fig. 1a). Since the discovery of the effect, extensive studies on MVE have made remarkable progress in the physics of spin fluctuation and electronic correlation in ferromagnetic metals[1].

Recently, a powerful tool for controlling spin fluctuation emerged in the field of spintronics: a spin current[7-12], a flow of spin angular momentum in a solid. By injecting a spin current into ferromagnets, magnetization fluctuation can be modulated via the angular momentum transfer between magnetization and a spin current[13,14]; as shown in Fig. 1b, when the injected spin current carries spins along the field direction (the $z$-direction), the spin current turns the magnetization **M** toward the $z$-direction via the spin-transfer torque[15], and the magnetization fluctuation is suppressed. By combining the effect with MVE, a fascinating hypothesis is made: volume of matter could be manipulated by applying a spin current: a spin current volume effect (SVE).

Here, we report an observation of SVEs in a $Tb_{0.3}Dy_{0.7}Fe_2$ film. $Tb_{0.3}Dy_{0.7}Fe_2$ is a typical ferromagnet exhibiting strong magnetostriction due to spin-lattice coupling[16-19]. We demonstrate the thickness modulation of the $Tb_{0.3}Dy_{0.7}Fe_2$ film by the spin current injection, a spin-current induced magnetostriction effect, which offers a way for magneto-mechanical control of mechanical actuators based on spintronics.

## Results

**Sample characterization and measurement setup.** Figure 1d shows a schematic illustration of the sample system used in the present study. To inject a spin current into a $Tb_{0.3}Dy_{0.7}Fe_2$ film, we used the spin Hall effect[20] (SHE) in a paramagnetic metal Pt. When a charge current, $j_c$, is applied to a Pt film, it is converted into a spin current, $j_s$, via the SHE, as shown in Fig. 1c. By putting a $Tb_{0.3}Dy_{0.7}Fe_2$ film onto a Pt film on a Si substrate by an electroplating method[21] (see Methods for details), the spin current with the spin polarization $\sigma \propto j_c \times n$ is injected into the $Tb_{0.3}Dy_{0.7}Fe_2$ film, where **n** is a normal vector to the interfacial plane. The magnetostriction coefficient of the $Tb_{0.3}Dy_{0.7}Fe_2$ film fabricated in the present study was found to be $\sim550 \times 10^{-6}$ around 1400 Oe[21] (see Supplementary Note 5 for details).

The thickness change of the film is measured by means of laser Doppler vibrometry[22] (LDV). In the measurement, as shown in Fig. 1d, an a.c. spin current is injected into the $Tb_{0.3}Dy_{0.7}Fe_2$ film by applying an a.c. electric current $j_c$ to the Pt film, where the spin-transfer torque $\propto j_c$ can increase and decrease the **M** fluctuation alternatively. The **M** fluctuation decrease (increase) causes the volume expansion (shrinkage) of the $Tb_{0.3}Dy_{0.7}Fe_2$ film. Due to the in-plane constraints of the film fabricated on the Si substrate, the volume change should accompany a thickness change of the film, as shown in Fig. 1c. The LDV detects the out-of-plane displacement of the film surface in response to the a.c. spin current injection in terms of the Doppler shift of light reflected at the surface of the film. To extract the displacement synchronized with the input a.c. spin current, we performed Fourier transform and obtained its frequency spectra (see Methods for details). All the measurements were performed at room temperature.

**Observation of spin current volume effect.** Figure 2a shows the obtained frequency $f$ spectrum of the vibrational amplitude $A$ and

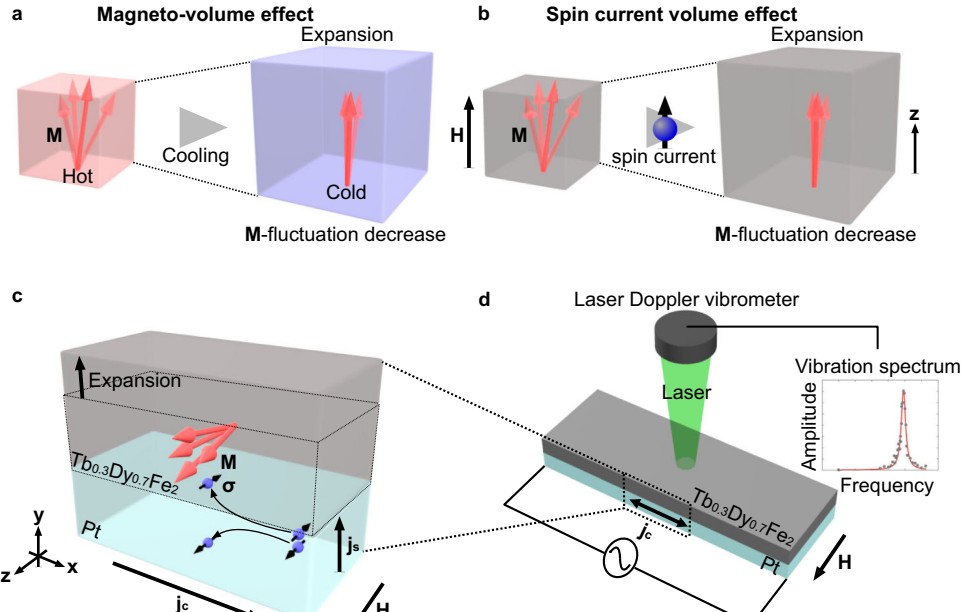

**Fig. 1 Concept of spin current volume effects (SVEs). a** A schematic illustration of the magneto-volume effect (MVE). A ferromagnet expands (shrinks) via the spin-lattice coupling when spin fluctuation in the magnet decreases (increases) due to the magnetic field application or temperature modulation. The left (right) panel shows the ferromagnet at higher (lower) temperature. **b** A schematic illustration of SVE. The volume of a ferromagnet can be tuned by spin current injection. The left (right) panel shows the ferromagnet before (after) the spin current injection. **c** A schematic illustration of SVE induced by the spin Hall effect (SHE) in a $Pt/Tb_{0.3}Dy_{0.7}Fe_2$ bilayer system. **H**, **M**, $j_c$, $j_s$, and $\sigma$ denote the magnetic field, magnetization of the $Tb_{0.3}Dy_{0.7}Fe_2$ film, a charge current, a spin current, and the spin polarization vector of $j_s$, respectively. When $j_c$ flows in the $+x$ direction in the Pt film, $j_s$ with $\sigma$ is injected into the $Tb_{0.3}Dy_{0.7}Fe_2$ film, and **M** fluctuation in the $Tb_{0.3}Dy_{0.7}Fe_2$ film decreases, causing the volume expansion via the spin-lattice coupling. The volume expansion should accompany a thickness change of the $Tb_{0.3}Dy_{0.7}Fe_2$ film due to the in-plane constraints of the film on a Si substrate. **d** A measurement setup in the present study. An a.c. spin current is injected into the $Tb_{0.3}Dy_{0.7}Fe_2$ film via the SHE by applying $j_c$ to the Pt film, and the mechanical vibrational spectrum for the sample surface is obtained by laser Doppler vibrometry.

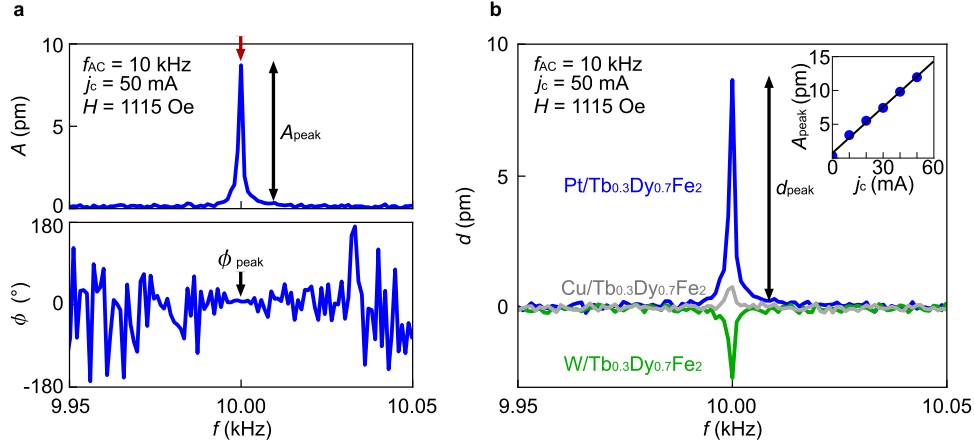

**Fig. 2 Observation of spin-current induced mechanical vibration in $Tb_{0.3}Dy_{0.7}Fe_2$. a** Frequency $f$ spectra of the mechanical vibration amplitude $A$ (upper panel) and the vibration phase $\phi$ (lower panel) for the $Pt/Tb_{0.3}Dy_{0.7}Fe_2$ sample at the a.c. current frequency $f_{AC} = 10$ kHz and the a.c. current amplitude $j_c = 50$ mA, measured by applying the magnetic field $H = 1115$ Oe. $A_{peak}$ and $\phi_{peak}$ are $A$ and $\phi$ at $f = f_{AC}$, respectively. **b** $f$ spectra of $d = A\cos\phi$ for the $Pt/Tb_{0.3}Dy_{0.7}Fe_2$ (a blue solid curve), $W/Tb_{0.3}Dy_{0.7}Fe_2$ (a green solid curve), and $Cu/Tb_{0.3}Dy_{0.7}Fe_2$ (a gray solid curve) samples. The inset shows the $j_c$ dependence of $A_{peak}$ at $H = 1530$ Oe, where the blue plots and the black solid curve are the measurement data and a linear fitting line, respectively.

phase $\phi$ for the $Pt/Tb_{0.3}Dy_{0.7}Fe_2$ sample. We applied $j_c$ at the frequency of $f_{AC} = 10$ kHz and the external magnetic field $H = 1115$ Oe along the $z$-axis. Importantly, as shown by the red arrow in Fig. 2a, a clear peak appears at $f = 10$ kHz ($= f_{AC}$) in the $A$ spectrum. The result means that the surface of the $Tb_{0.3}Dy_{0.7}Fe_2$ film mechanically vibrates at the same frequency as that of the a.c. current. The $j_c$ dependence of $A$ at $f = f_{AC}$, $A_{peak}$, also indicates that the amplitude of the mechanical vibration is proportional to $j_c$ (see the inset to Fig. 2b). The results exclude the thermal expansion due to the Joule heating ($\propto j_c^2$), whose frequency is $2f_{AC}$ (see Supplementary Note 2). We also confirmed that the observed peak exhibits the frequency shift in response to the change in the $f_{AC}$ value (see Supplementary Note 3).

To clarify the origin of the observed mechanical vibration, we carried out control experiments by replacing Pt with paramagnetic metals W and Cu, where the sign of the spin-Hall angle in W is opposite to that in Pt[23,24], while Cu exhibits minute SHEs[23]. As shown in Fig. 2b, the signed amplitude of the vibration $d \equiv A\cos\phi$ for a $W/Tb_{0.3}Dy_{0.7}Fe_2$ sample exhibits an opposite sign to that for the $Pt/Tb_{0.3}Dy_{0.7}Fe_2$ sample. This sign reversal is consistent with the signs of the spin-transfer torque created by Pt and W on $Tb_{0.3}Dy_{0.7}Fe_2$. We also performed a similar experiment for a $Cu/Tb_{0.3}Dy_{0.7}Fe_2$ sample. In the sample, the peak of $d$ at $f = f_{AC}$ is much smaller than that in the $Pt/Tb_{0.3}Dy_{0.7}Fe_2$ sample (a gray solid curve in Fig. 2b). The results indicate that the observed mechanical vibration is due to the spin current injection via the SHE.

Figure 3 shows the magnetic field $H$ dependence of the mechanical vibration. In the inset of Fig. 3b, we show the magnetization curve of the $Pt/Tb_{0.3}Dy_{0.7}Fe_2$ sample measured with changing $H$ from $-1700$ Oe to $1700$ Oe (the maximum field we can apply in our LDV system) along the $z$-direction. In this $H$ range (much less than the magnetization saturation field of the $Tb_{0.3}Dy_{0.7}Fe_2$ film $\sim 7000$ Oe), **M** is not fully saturated along the field direction. To compare the mechanical vibration signal with the magnetization curve, we show the $A$ and $\phi$ spectra at each $H$ for the $Pt/Tb_{0.3}Dy_{0.7}Fe_2$ sample (see Fig. 3c). Here, $\phi$ at $f = f_{AC}$, $\phi_{peak}$, is shifted by 180° between $H = 1410$ Oe and $H = -1400$ Oe, showing that the sign of the vibration displacement is reversed by reversing the field direction. In the entire field range, $A_{peak}$ increases with $|H|$, while $\phi_{peak}$ changes by 180° at around $H = 0$ (blue plots in Fig. 3a). In Fig. 3b, we plot the $H$ dependence of the singed amplitude of the vibration

$d_{peak} = A_{peak}\cos\phi_{peak}$. $d_{peak}$ increases with the increase of $H$, which aligns **M** along with the field direction. Furthermore, as shown in Fig. 3a, b, d, e, the $W/Tb_{0.3}Dy_{0.7}Fe_2$ sample exhibits a clear sign reversal of the displacement (green plots and curves) before the possible oxidization of the W film and the $Cu/Tb_{0.3}Dy_{0.7}Fe_2$ sample shows suppression of the displacement (gray plots and curves), being consistent with the characteristics of the spin current injection via the SHE. The results imply that the observed mechanical vibration originates from the interaction between **M** and spin currents.

## Comparison between experiments and theoretical model for spin current volume effect. 
Now we discuss the origin of the observed mechanical displacement due to the spin current injection. The injected spin current interacts with **m**, partial magnetization responsible for the volume effect, and exerts the spin-transfer torque[15] $\tau_{stt} \propto \mathbf{m} \times (\mathbf{m} \times \sigma)$ on **m**. The frequency of the current 10 kHz is much less than that of the magnetization dynamics $\sim$GHz, and the effective damping and fluctuation of **m** are modulated by $\tau_{stt}$ via the anti-damping spin torque (Slonczewski spin torque[15]) mechanism; $\tau_{stt}$ suppresses (enhances) the **m** thermal fluctuation and increases (decreases) the thermally averaged **m** intensity when $\sigma$ is antiparallel (parallel) to **m**. The **m** intensity increase (decrease) induces the expansion (shrinkage) of the $Tb_{0.3}Dy_{0.7}Fe_2$ film via the spin-lattice coupling (see Fig. 4b), resulting in the out-of-plane mechanical displacement. In contrast, when $\sigma$ is perpendicular to **m**, the **m** fluctuation remains unchanged due to the cancellation of $\tau_{stt}$, and the volume change does not occur. We note that the effect of the anti-damping torque is maximized when $\mathbf{m} \parallel \sigma$ because it is exerted on the **m** fluctuation component[7]. Based on the above scenario, we constructed a theoretical model for the SVE. The magnetization dynamics under thermal fluctuation is calculated from the stochastic LLG equation[25] $\frac{d\mathbf{m}}{dt} = -\gamma\mathbf{m} \times [\mathbf{H} + \mathbf{h}(t)] + \frac{\alpha}{m_s}\mathbf{m} \times \frac{d\mathbf{m}}{dt} + \tau_{stt}$, where $\gamma$, $\alpha$, and $m_s$ are the gyromagnetic ratio, the magnetic damping coefficient, and the saturation magnetization, respectively. The thermal fluctuation of **m** is taken into account with the random magnetic field $\mathbf{h}(t)$. By combining the above equation, the fluctuation-dissipation theorem, and phenomenological magnetoelastic theory[26], we derived the mechanical displacement $d_{SVE}$ due to the SVE in the linear response to $j_s$ (see Supplementary

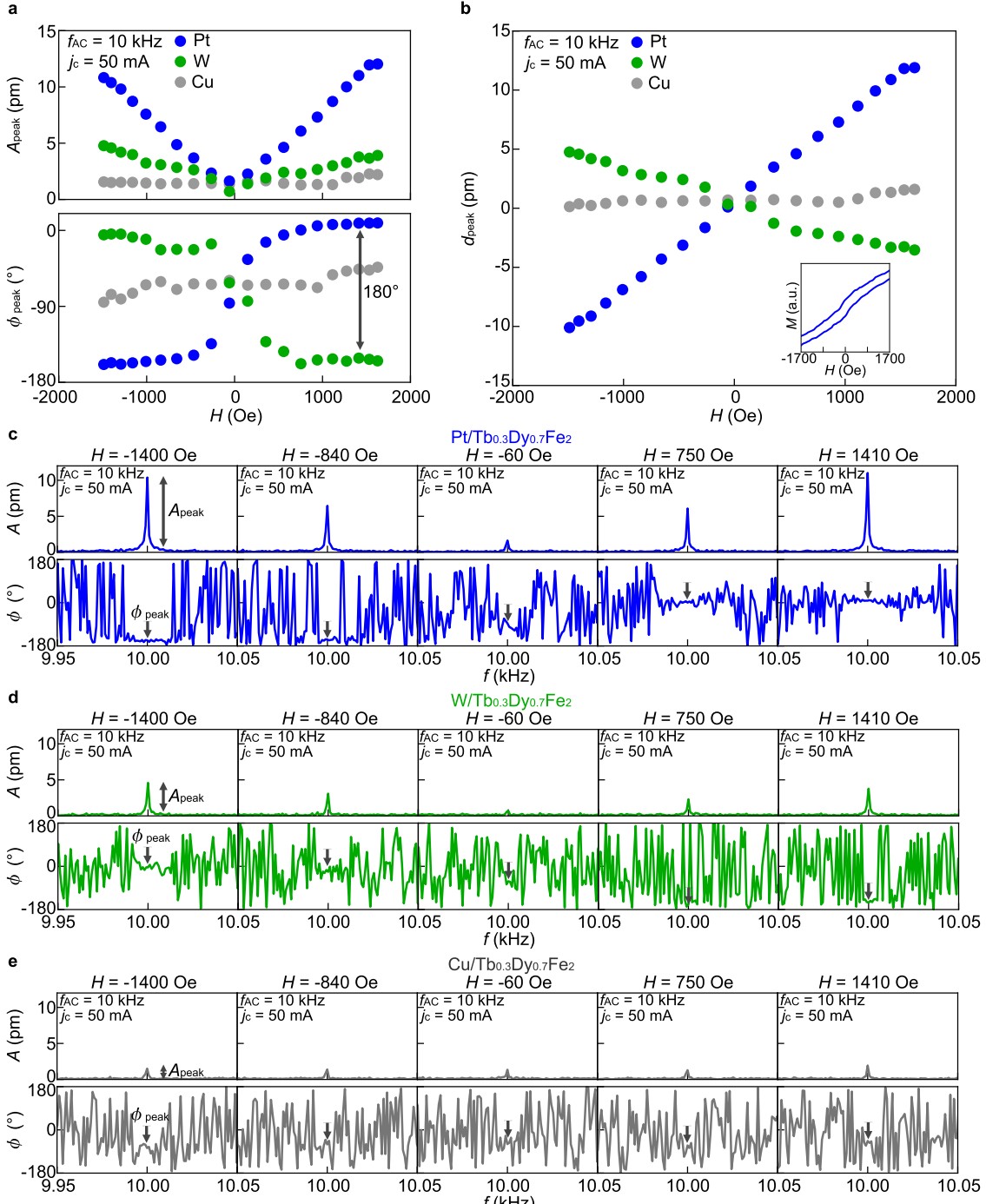

**Fig. 3 Magnetic field dependence of spin-current induced mechanical vibration. a** The $H$ dependence of $A_{peak}$ (upper panel) and $\phi_{peak}$ (lower panel) for the $Pt/Tb_{0.3}Dy_{0.7}Fe_2$ (blue plots), $W/Tb_{0.3}Dy_{0.7}Fe_2$ (green plots), and $Cu/Tb_{0.3}Dy_{0.7}Fe_2$ (gray plots) samples. **b** The $H$ dependence of $d_{peak} = A_{peak}\cos\phi_{peak}$ for the $Pt/Tb_{0.3}Dy_{0.7}Fe_2$ (blue plots), $W/Tb_{0.3}Dy_{0.7}Fe_2$ (green plots), and $Cu/Tb_{0.3}Dy_{0.7}Fe_2$ (gray plots) samples. The inset shows the $H$ dependence of the magnetization $M$ for the $Pt/Tb_{0.3}Dy_{0.7}Fe_2$ sample. **c–e** $f$ spectra of $A$ (upper panel) and $\phi$ (lower panel) at each $H$ for the $Pt/Tb_{0.3}Dy_{0.7}Fe_2$ (**c**), $W/Tb_{0.3}Dy_{0.7}Fe_2$ (**d**), and $Cu/Tb_{0.3}Dy_{0.7}Fe_2$ (**e**) samples. The values of $f_{AC}$ and $j_c$ are 10 kHz and 50 mA, respectively.

Note 7 and 8 for details):

$$d_{SVE} = \frac{ak_BT}{\alpha m_s H^2 V}\left(m_s - \frac{k_BT}{HV}\right)j_s\sin\theta, \qquad (1)$$

where $k_B$, $T$, $V$, $a$, and $\theta$ are the Boltzmann constant, temperature, the magnetic coherence volume[25] of the $Tb_{0.3}Dy_{0.7}Fe_2$ film, a known constant parameter, and the relative angle between **H** and $\mathbf{j}_c$ (see Fig. 4a), respectively. Here, the result gives us the $\theta$ dependence of the SVE: $d_{SVE}$ is proportional to $\sin\theta$. When the

external field is much weaker than the magnetization saturation field, $d_{SVE} \propto \sin\theta$, where $H$ in Eq. (1) is replaced with the internal magnetic field in each magnetic domain and $\theta$ represents the relative angle between local magnetization in each domain and $\mathbf{j}_c$. The $\sin\theta$ averaged over the magnetic domains increases with the external magnetic field application, consistent with the observed $H$ dependence of $d_{peak}$ (see also Supplementary Note 8).

Figure 4c shows the $\theta$ dependence of the measured $d_{peak}$. We found that $d_{peak}$ exhibits a clear $\sin\theta$ dependence for both the $Pt/$

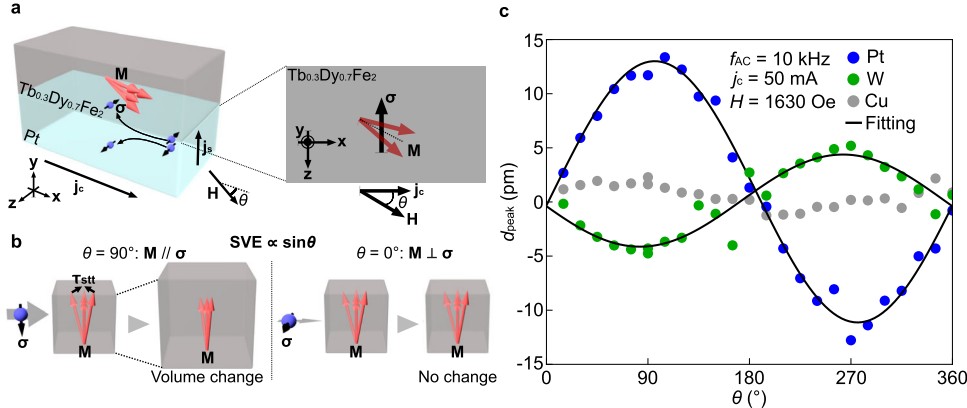

**Fig. 4 Field-direction dependence of spin-current induced mechanical vibration. a** A schematic illustration of the field-direction dependence measurement. $\theta$ denotes the relative angle between **H** and $\mathbf{j_c}$. **b** A schematic illustration of the SVE when **M** || **σ** (left panel) and **M** ⊥ **σ** (right panel). The spin-transfer torque $\tau_{stt} \propto \mathbf{M} \times (\mathbf{M} \times \sigma)$ decreases (increases) the **M** fluctuation when **σ** is antiparallel (parallel) to **M**, causing the volume expansion (shrinkage) of the $Tb_{0.3}Dy_{0.7}Fe_2$ film. **c** The $\theta$ dependence of $d_{peak}$ for the Pt/$Tb_{0.3}Dy_{0.7}Fe_2$ (blue plots), W/$Tb_{0.3}Dy_{0.7}Fe_2$ (green plots), and Cu/$Tb_{0.3}Dy_{0.7}Fe_2$ (gray plots) samples. The black solid curves show the theoretical fitting result. The values of $f_{AC}$, $j_c$, and $H$ were set to 10 kHz, 50 mA, and 1630 Oe, respectively.

$Tb_{0.3}Dy_{0.7}Fe_2$ (blue plots) and W/$Tb_{0.3}Dy_{0.7}Fe_2$ (green plots) samples. The result is consistent with the theoretically obtained $d_{SVE} \propto \sin\theta$ and rules out the spin-current induced shear magnetostriction $\propto \cos\theta$, which originates from the magnetization rotation due to $\tau_{stt}$ (see Supplementary Note 8 for details). The agreement between the experimental results and the theoretical calculation supports our interpretation that the observed mechanical vibration is attributed to the volume change due to the SVE.

## Discussion

Here, we discuss the influence of other effects on the observed mechanical displacements in the (Pt, W, and Cu)/$Tb_{0.3}Dy_{0.7}Fe_2$ samples. The first one is Lorentz force due to the a.c. current under the magnetic fields. By carrying out control experiments using (Pt, W, and Cu)/Si samples without the $Tb_{0.3}Dy_{0.7}Fe_2$ layer, we found that the mechanical peak signal disappears in the absence of the $Tb_{0.3}Dy_{0.7}Fe_2$ layer (see Supplementary Note 1). The result implies that the Lorentz force is irrelevant to the observed mechanical effect. We also examined the Oersted field effect due to the a.c. current flowing in the paramagnetic metals which might induce magnetostriction of the $Tb_{0.3}Dy_{0.7}Fe_2$ film, but we found that this cannot be responsible for the observed paramagnetic metal dependence (sign reversal between Pt/$Tb_{0.3}Dy_{0.7}Fe_2$ and W/$Tb_{0.3}Dy_{0.7}Fe_2$), although the small mechanical signal in the Cu/$Tb_{0.3}Dy_{0.7}Fe_2$ sample might be attributed to such Oersted field effects or a small finite SHE in the Cu film[27–29].

In summary, we found spin current volume effects (SVEs), volume modulation by spin current injection, in $Tb_{0.3}Dy_{0.7}Fe_2$ films. The SVE observed here enables the direct mechanical actuation of a magnetostrictive thin film by using a spin current, which can be applied to making mechanical actuators driven by spin currents free from electricity. The high controllability of the SVE in terms of magnetic fields will present great advantages in designing spintronics-based mechanical devices.

## Methods

**Sample preparation.** We used an electroplating method[21] to grow the $Tb_{0.3}Dy_{0.7}Fe_2$ film on the paramagnetic metals (see Supplementary Note 5 for details). The polycrystalline $Tb_{0.3}Dy_{0.7}Fe_2$ film with the thickness of ~100 nm was fabricated on the Pt film with the thickness of 140 nm (the W and Cu films with the thickness of 100 nm), which was sputtered as a seed electrode for electroplating on a Si substrate with the size of 20 mm × 20 mm. The obtained samples were cut into

2 mm wide and 10 mm long pieces. The resistance of the $Tb_{0.3}Dy_{0.7}Fe_2$ film is in the order of 0.1 MΩ while the resistance of the paramagnetic films is less than 10 Ω. Therefore, when an electric current flows in the (Pt, W, and Cu)/$Tb_{0.3}Dy_{0.7}Fe_2$ samples, the portion of the electric current in the paramagnetic metal films is much greater than that in the $Tb_{0.3}Dy_{0.7}Fe_2$ film. The spin current is injected into the $Tb_{0.3}Dy_{0.7}Fe_2$ film via the SHE in the paramagnetic films and it modulates magnetization fluctuation (a type of the reverse processes of the dynamic spin pumping[7,13]). In electroplating methods, it is not possible to fabricate a single layer of $Tb_{0.3}Dy_{0.7}Fe_2$ without the seed electrode films, such as Pt.

**Mechanical vibration measurement setup.** The samples were fixed with varnish on a stage located between the magnetic poles of an electromagnet. An a.c. charge current was applied to the samples to induce the SVE. The mechanical vibration of the sample surface was measured by means of LDV, where a laser light with the wavelength $\lambda = 532$ nm was split into a reference beam and an incident beam. The incident laser beam was focused on the $Tb_{0.3}Dy_{0.7}Fe_2$ film surface. The reflected light from the sample surface was analyzed with an LDV system (MSA-100-3D, Polytec, Inc.) to obtain the displacement and velocity of the surface along the laser-beam direction as a function of time. The data were Fourier transformed into $f$ spectra of $A$ and $\phi$. All the measurements were performed at room temperature and in a high vacuum of ~$10^{-4}$ Pa.

## Data availability

The data that support the findings of this study are available from the corresponding author upon reasonable request.

## Code availability

The codes used in theoretical calculations are available from the corresponding author upon reasonable request.

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

## Acknowledgements

This work was supported by ERATO "Spin Quantum Rectification Project" (No. JPMJER1402) from JST, Japan, CREST (Nos. JPMJCR20C1 and JPMJCR20T2) from JST, Japan, Grant-in-Aid for Scientific Research (S) (No. JP19H05600), Grant-in-Aid for Scientific Research (B) (No. JP20H02599) from JSPS KAKENHI, Japan, NEC Corporation, and Institute for AI and Beyond of the University of Tokyo. H.A. is supported by JSPS through a research fellowship for young scientists (No. JP20J21622) and GP-Spin at Tohoku University. X-ray photoelectron spectroscopy measurements were conducted at the Advanced Characterization Nanotechnology Platform, the University of Tokyo, supported by MEXT, Japan.

## Author contributions

H.A. and H.S. contributed equally to this work. E.S. supervised the study. H.S. and T.O. prepared samples. H.A. carried out the experiments and analyzed the data with help from S.D., T.K., and Y.O. H.A. formulated the theoretical model with help from S.T. H.A. and E.S. prepared the manuscript. All the authors discussed the results and commented on the manuscript.

## Competing interests

The authors declare no competing interests.
