## [Peer Review File · Nature Communications]

Observation of spin-current striction in a magnetREVIEWER COMMENTS

Reviewer #1 (Remarks to the Author):

I have reviewed the paper focusing on Terfenol-D thin film actuated using the spin hall effect and its related influence on changes in volume. While I do find the paper interesting and I am not aware of other papers experimentally actuating TD with SHE I find the paper is lacking sufficient details to fully understand and appreciate the work. See below for comments associated with this statement.

1. Authors do not provide any information on how the films was made, what the films properties are, and ensuring the film is of good quality. Things like crystal versus amorphous and thickness are not present and these prevent this work from being assessed or repeated by other researchers.
2. This reviewer believes that the response measured by the authors is magnetostriction induced by SHE. This could be a novel contribution if authors were to provide background on if this is the first to demonstrate this on Terfenol-D. However, the correlation with the papers motivation that is controlling thermal expansion effects doe not appear to be present. Thus, this reviewer perceives this paper more about can you actuate thin film TD with SHE rather than the next extrapolation to controlling volume changes. Thus, this reviewer doe not fully appreciate the point the authors are trying to make.
3. While the authors have moderately convinced this reviewer that they are looking at magnetostriction, without some estimates of what these could be as well as figure 2b showing a linear magnetostriction trend without saturation raises concerns.
4. This reviewer encourages the authors to provide more details if necessary in the supplemental section regarding the comments above but even with these it may not change this reviewers opinion. That is, important information is missing which is necessary for this review to develop a full opinion on this work.

Reviewer #2 (Remarks to the Author):

Arisawa et al. reported that the volume of a ferromagnetic Tb_{0.3}Dy_{0.7}Fe₂ can be manipulated by the spin current injection, and named it as spin current volume effect. The main result is that they compared the volume change using different normal metallic layers, including Pt, W, and Cu. They observed the opposite trends between Pt and W with respect to the charge current, and claimed that this behavior was due to the opposite signs of the spin Hall angle between Pt and W. If the statement of spin current volume effect is true, it could be interesting for the field of spin-mechanics. However, based on the experimental evidences in the present manuscript, I find it not convincing enough for such a strong conclusion. Hence, it is not recommended for publication unless the authors provide more convincing evidences to fully support their statement.

- 1) In this manuscript, the authors directly jump to the spin current volume effect without the experimental demonstration of the magneto-volume effect of Tb_{0.3}Dy_{0.7}Fe₂.

Since the whole work is based on the spin-fluctuation induced volume change, the basic characterization of the conventional magneto-volume effect of Tb_{0.3}Dy_{0.7}Fe₂ is necessary for the conclusion of spin current volume effect.

- 2) Following #1), the authors should do a quantitative comparison of the spin current volume

effect and the magneto-volume effect. Since the spin current injection only modifies the magnetization a little, and the effect can be estimated based on how much the magnetization changes. This quantitative comparison can be a further support to the conclusion of this manuscript.

3) Also following #1), what is the optimistic temperature for the largest magneto-volume effect? Around room temperature, does this effect increase as temperature increases?

Since the authors performed the current dependence up to a relative large current of 50 mA (Fig. 2b inset), the sample temperature is expected to increase. How does this affect the magneto-volume effect and the spin current volume effect as the author named?

4) The authors selected the supportive results, the opposite spin Hall angle signs of Pt and W, to be mainly discussed in this manuscript. However, the other aspects of the results do not seem to support their conclusion.

For example, the magnetic field dependence in Fig. 3 does not agree with this scenario. When the external magnetic field is larger, the effective magnetization is larger, the same amount of spin current gives rise to less effect for a larger amount of magnetic moment ($\sim 1/M_{\text{eff}}$). As a result, a smaller effect is expected at a large magnetic field. However, this seems to be contradictive to the observation in this work.

Besides, there is a hysteresis of the magnetization of Tb_{0.3}Dy_{0.7}Fe₂ as a function of the magnetic field, and a net magnetization is expected at zero magnetic field. Hence, such a spin volume effect is expected at zero magnetic field, and the volume change ratio expected to follow the magnetization curves as a function of magnetic field. However, these expectations do not agree with the experimental results in this work.

5) The authors observed the opposite trends between Pt and W with respect to the charge current. However, if one looks carefully, there was already a strong signal of Cu/ Tb_{0.3}Dy_{0.7}Fe₂ as shown in Fig. 2b. As the spin Hall angle of Cu is negligible, no effect is expected.

Furthermore, how about the control sample of a single layer Tb_{0.3}Dy_{0.7}Fe₂? This control experimental would be necessary to further support their claim that the spin current was the major cause.

6) Why did the authors select the current frequency to be 10 kHz for their experiment? Did the authors also perform the experiment using other frequencies from low frequency close to the DC regime to high frequencies up to GHz?

7) It is known that the magnetization dynamics of a ferromagnetic materials is in the \sim GHz range, and the spin lifetimes in a typical ferromagnet are usually less than 1 ps. Hence, it is expected the injected spin current will be fully relaxed into the ferromagnet in the 10 KHz regime. As shown in previous spin-orbit torque ferromagnetic resonance measurement, the affected magnetization dynamics due to the spin-Hall torque in Pt/FM bilayer systems also happens in the GHz range.

Following this, does the authors expect the volume change by the 10 kHz AC spin current injection?

8) As the authors claimed that the spin transfer torque is $\sim \mathbf{m} \times (\mathbf{m} \times \boldsymbol{\sigma})$, hence, a zero spin torque is expected if all the \mathbf{m} moments of Tb_{0.3}Dy_{0.7}Fe₂ are parallel to $\boldsymbol{\sigma}$, the spin current direction from Pt.

However, in this work, the authors claimed that the maximum effect is observed when \mathbf{m} is parallel to $\boldsymbol{\sigma}$. This is contrary to the common understanding of spin-transfer torque.

More likely, the torque is applied on the fluctuation parts of \mathbf{m} moments that are perpendicular to the external magnetic field direction. The explanation of the statement is suggested to be revised for general audience.

9) Does the expansion/suppression of the volume happen along the magnetization direction? In Fig. 1c, the expansion was noted to be perpendicular to the magnetization direction, but in Fig. 1b, it seems that the expansion is the same as the magnetization direction.

Reviewer #3 (Remarks to the Author):

This article describes the volume change of $\text{Tb}(0.3)\text{Dy}(0.7)\text{Fe}_2$ under dynamic spin injection by the spin Hall effect in a platinum strip attached to the sample. The experiment is a tour-de-force: the change in size of the sample is of the order of picometers, and is measured by a laser Doppler vibrometer.

I am not quite convinced of the validity of all the experimental claims. Here are a few questions I think the authors should address.

1. Following is my main concern: the current is an AC current. In an applied external magnetic field, this generates a mechanical force in the system at the exact same frequency as the excitation. This mechanical force can induce a motion of the sample that the system will pick up, given its superb pm-level sensitivity. The motion can be amplified by a mechanical resonance, giving rise to the resonant vibrations picked up by the detector. Basically, I wonder if it is not the whole setup that vibrates, rather than the $\text{Tb}(0.3)\text{Dy}(0.7)\text{Fe}_2$ that changes thickness. The resistivity of Pt, W and Cu being different, the effects will be different for the different metals.
2. The experiment really purports to measure a change in dimension of the sample in the direction perpendicular to the film plane, so it strictly is a measurement of a linear expansion, not a volume expansion.
3. Is there a difference between this and magnetostriction, an effect well known to exist in the presence of a static magnetic field? Is this not equivalent to magnetostriction under dynamic spin pumping?
4. Is the absence of an effect when the Pt strip is replaced by a Cu one sufficient to conclude that the effect is not due to magnetostriction in the current's Oersted field? An alternative explanation for the data would then be that, with a Pt film, most of the current flows in the $\text{Tb}(0.3)\text{Dy}(0.7)\text{Fe}_2$ whereas with the Cu film, most of the current flows in the Cu. The Oersted fields perceived by the $\text{Tb}(0.3)\text{Dy}(0.7)\text{Fe}_2$ will then be different and therefore the magnetostriction of the $\text{Tb}(0.3)\text{Dy}(0.7)\text{Fe}_2$ will also be.
5. Magnetostriction is even in field. Fig 3b gives the signal as a function of the bias field: the signal is zero without a bias field, and an odd function of that bias field. The presence of a background field in the experiment creates an offset in the strain/field relation, and either the spin current or the Oersted field induce a variation in field around that offset point, which then will give a response that is odd in field. Therefore I am not sure I can uniquely attribute the result to spin injection.
5. There is also electrostriction, typically very large only in ferroelectrics, but non-zero even in electrical conductors. Given the sensitivity of these measurements, how do the authors know it is not this that is measured. Pt, W and Cu probably have very different electrostriction coefficients.

Reply to the Reviewer #1

We thank the reviewer for the valuable comments on our manuscripts and for finding our experimental results interesting. Following the comments, we carefully revised the manuscripts as shown in the following.

(Reviewer's comment/question 1-1)

1. Authors do not provide any information on how the films was made, what the films properties are, and ensuring the film is of good quality. Things like crystal versus amorphous and thickness are not present and these prevent this work from being assessed or repeated by other researchers.

(Authors' response 1-1)

We thank the reviewer for the comment. We used an electroplating method [Shim, H. *et al.*, *Micromachines* **11**, 523 (2020)] to grow the polycrystalline $Tb_{0.3}Dy_{0.7}Fe_2$ film by using the paramagnetic metal (Pt, W, and Cu) layers as electrodes for plating, as described in the Methods section in our previous manuscript. In the revised manuscript, we mentioned the method in the main text, too (Lines 39-40, Page 2).

We apologize for the lack of information on the properties of the $Tb_{0.3}Dy_{0.7}Fe_2$ film. We show the magnetostriction data for the $Tb_{0.3}Dy_{0.7}Fe_2$ film in Fig. R1; the $Tb_{0.3}Dy_{0.7}Fe_2$ film was found to exhibit the magnetostriction coefficient $\sim 1200 \times 10^{-6}$ around the magnetization saturation, which is the highest value among those previously reported for $Tb_{0.3}Dy_{0.7}Fe_2$ films and comparable to the bulk value. We used the same film fabrication method optimized in the previous study [Shim, H. *et al.*, *Micromachines* **11**, 523 (2020)]. Following the reviewer's comment, in the revised manuscript, we added a sentence on the sample properties to the main text (Lines 42-43, Page 2).

Fig. R1 | Magnetic field dependence of magnetostriction coefficient of $Tb_{0.3}Dy_{0.7}Fe_2$ film [Shim, H. *et al.*, *Micromachines* **11, 523 (2020)].**

(Reviewer's comment/question 1-2.1)

2. This reviewer believes that the response measured by the authors is magnetostriction induced by SHE. This could be a novel contribution if authors were to provide background on if this is the first to demonstrate this on Terfenol-D.

(Authors' response 1-2.1)

Thank you for the comment. As the reviewer pointed out, the actuation of the $Tb_{0.3}Dy_{0.7}Fe_2$ films via the SHE, to the best of our knowledge, has never been reported before. Following the suggestion, we added a sentence on the background of our study to the main text (Lines 32-34, Page 2).

(Reviewer's comment/question 1-2.2)

However, the correlation with the papers motivation that is controlling thermal expansion effects does not appear to be present. Thus, this reviewer perceives this paper more about can you actuate thin film TD with SHE rather than the next extrapolation to controlling volume changes. Thus, this reviewer does not fully appreciate the point the authors are trying to make.

(Authors' response 1-2.2)

We apologize for the unclear explanation. The motivation of the study is not the control of thermal expansion, but it is to control the magneto-volume effect using a spin current. The magneto-volume effect is one of the broad magnetostriction effects. In thin films on a substrate, dominant magnetostriction is inevitably accompanied by volume changes (thickness changes) due to the in-plane constraints from the substrate. Especially when the field is parallel to the magnetization, magnetization fluctuation has been studied as an important mechanism for the volume changes induced by fields. As the reviewer pointed out, the demonstration of the spin-current induced magnetostriction will present a new way to the spintronics-based actuation of a thin magnetostrictive film. To make the point clear, following the reviewer's comment, we rephrased the title of the paper (Lines 1, Page 1) and the sentences on the motivation in the main text (Lines 10-12, Page 1 and Lines 132-133, Page 5).

(Reviewer's comment/question 1-3)

3. While the authors have moderately convinced this reviewer that they are looking at magnetostriction, without some estimates of what these could be as well as figure 2b showing a linear magnetostriction trend without saturation raises concerns.

(Authors' response 1-3)

Thank you for the comment. Following the reviewer's suggestion, we carried out quantitative analysis on the spin current volume effect in the present system by combining our theoretical analysis and the magnetostriction coefficient of the $\text{Tb}_{0.3}\text{Dy}_{0.7}\text{Fe}_2$ film shown in Fig. R1 [please see the revised Supplementary Information (Lines 147-159, Page 11) for details]. As for the field dependence, the saturation magnetic field of the present $\text{Tb}_{0.3}\text{Dy}_{0.7}\text{Fe}_2$ film is too high (~ 7 kOe) to be applied by using our laser Doppler vibrometer system. In the revised manuscript, we addressed this point in the main text (Lines 73-74, Page 3).

(Reviewer's comment/question 1-4)

4. This reviewer encourages the authors to provide more details if necessary in the supplemental section regarding the comments above.

(Authors' response 1-4)

We thank the reviewer for the comment. Following the reviewer's suggestions, we added sentences on the sample properties, the motivation and the novelty of our study, and the estimation of the SHE-induced mechanical displacement to the revised manuscripts.

[Reply to the reviewer #2]

We thank the reviewer for the valuable comments on our manuscripts. Following the comments, we carefully revised the manuscripts as shown below.

(Reviewer's comment/question 2-1)

1) In this manuscript, the authors directly jump to the spin current volume effect without the experimental demonstration of the magneto-volume effect of Tb_{0.3}Dy_{0.7}Fe₂.

Since the whole work is based on the spin-fluctuation induced volume change, the basic characterization of the conventional magneto-volume effect of Tb_{0.3}Dy_{0.7}Fe₂ is necessary for the conclusion of spin current volume effect.

(Authors' response 2-1)

We thank the reviewer for the comment. As shown in Fig. R2, in the previous study [Shim, H. *et al.*, *Micromachines* **11**, 523 (2020)], we reported the magneto-volume change for the present Tb_{0.3}Dy_{0.7}Fe₂ film and estimated the magnetostriction coefficient to be about 1200×10^{-6} around the magnetization saturation. Following the reviewer's suggestion, we addressed the data in the main text (Lines 42-43, Page 2).

Fig. R2 | Magnetic field dependence of magnetostriction coefficient of Tb_{0.3}Dy_{0.7}Fe₂ film [Shim, H. *et al.*, *Micromachines* **11, 523 (2020)].**

(Reviewer's comment/question 2-2)

2) Following #1), the authors should do a quantitative comparison of the spin current volume effect and the magneto-volume effect. Since the spin current injection only modifies the magnetization a little, and the effect can be estimated based on how much the magnetization changes. This quantitative comparison can be a further support to the conclusion of this manuscript.

(Authors' response 2-2)

Thank you for the comment. By combining our theoretical analysis (please see the Supplementary Information) and the magnetostriction coefficient of the $\text{Tb}_{0.3}\text{Dy}_{0.7}\text{Fe}_2$ film shown in Fig. R2, we carried out quantitative analysis on the spin current volume effect in the present system. From theoretical fitting to the experimental data, we estimated an internal magnetic field acting on the rare earth ions that are responsible for magnetostriction of $\text{Tb}_{0.3}\text{Dy}_{0.7}\text{Fe}_2$ [Clark, A. E., AIP Conf. Proc. **18**, 1015 (1974), Jiles, D. C., J. Phys. D: Appl. Phys. **27**, 1 (1994), and Ren, W. J. *et al.*, Chin. Phys. B **22**, 077507 (2013).] (please see the revised Supplementary Information for details). Following the reviewer's instruction, we added sentences on the calculation to the Supplementary Information (Lines 147-159, Page 11).

(Reviewer's comment/question 2-3.1)

3) Also following #1), what is the optimistic temperature for the largest magneto-volume effect? Around room temperature, does this effect increase as temperature increases?

(Authors' response 2-3.1)

We thank the reviewer for the question. The optimal temperature for the magneto-volume effect can be expected to be around 250 K, where $\text{Tb}_{0.3}\text{Dy}_{0.7}\text{Fe}_2$ is known to exhibit structural phase transition [Richard, B. Jr. *et al.*, Phys. Rev. Lett. **111**, 017203 (2013)] and the spin-lattice coupling can be enhanced. Therefore, when temperature increases from room temperature, the magnitude of the magneto-volume changes may decrease with decreasing magnetization. However, in the case of $\text{Tb}_{0.3}\text{Dy}_{0.7}\text{Fe}_2$, the magnetic phase transition temperature (Curie temperature) > 600 K is far above room temperature, and the magneto-volume change does not strongly depend on temperature around room temperature [Oomi, G. *et al.*, J. Magn. Soc. Japan **23**, 450-452 (1999)]; sensitive detection of the temperature dependence is difficult by the present laser Doppler system. In the revised Supplementary Information, we added a sentence on this point (Lines 41-42, Page 4).

(Reviewer's comment/question 2-3.2)

Since the authors performed the current dependence up to a relative large current of 50 mA (Fig. 2b inset), the sample temperature is expected to increase. How does this affect the magneto-volume effect and the spin current volume effect as the author named?

(Authors' response 2-3.2)

We appreciate the reviewer's comment. To examine the heating effect, we measured the a.c. current amplitude j_c dependence of the temperature increase ΔT for the Pt/Tb_{0.3}Dy_{0.7}Fe₂ sample surface by using a thermographic camera. As shown in Fig. R3, ΔT increases ($\propto j_c^2$) with the increase of j_c , and we found that $\Delta T \sim 0.15$ K at $j_c = 50$ mA, which is negligibly small compared to the system temperature in our experiment ~ 300 K. Such a small ΔT little changes the magneto-volume effect of Tb_{0.3}Dy_{0.7}Fe₂ around the room temperature [Oomi, G. *et al.*, J. Magn. Soc. Japan **23**, 450-452 (1999)]. Following the reviewer's comment, to make this point clear, we added a sentence to the main text (Lines 60-61, Page 3) and a section to the Supplementary Information (Lines 36-47, Page 4).

Fig. R3 | A.c. current dependence of sample surface temperature. The a.c. current amplitude j_c dependence of the temperature increase ΔT of the Pt/Tb_{0.3}Dy_{0.7}Fe₂ sample surface (black dots). A blue solid curve is a square fitting. The a.c. current frequency f_{AC} was set to 10 kHz.

(Reviewer's comment/question 2-4.1)

4) The authors selected the supportive results, the opposite spin Hall angle signs of Pt and W, to be mainly discussed in this manuscript. However, the other aspects of the results do not seem to support their conclusion.

For example, the magnetic field dependence in Fig. 3 does not agree with this scenario. When the external magnetic field is larger, the effective magnetization is larger, the same amount of spin current gives rise to less effect for a larger amount of magnetic moment ($\sim 1/M_{eff}$).

(Authors' response 2-4.1)

We thank the reviewer for the comment. As pointed out by the reviewer, field induced suppression of the spin current volume effect may appear due to the field induced increase of magnetization. However, in fact, it may appear when the field is much stronger: the typical condition for the suppression to appear is $\mu_B H \gg k_B T$ (μ_B , H , and T are the Bohr magneton, magnetic field, and temperature, respectively). When $H \lesssim k_B T / \mu_B$, the effect of the field application is just rotating magnetization in

each magnetic domain. Note that the anti-damping spin torque (Slonczewski spin torque) is rather maximized when the field is parallel to the local macroscopic magnetization, being consistent with the present experimental result. According to our theoretical calculation shown in the Supplementary Information, the displacement due to the spin current volume effect d_{SVE} is nearly independent of the magnitude of the “effective magnetization” M_{eff} . d_{SVE} depends only on $\cos\psi$, where ψ is the relative angle between the spin-current polarization and local magnetization. To make this point clear, following the reviewer’s comment, we added some sentences to the main text (Lines 108-113, Page 4) and the Supplementary Information (Lines 147-159, Page 11).

(Reviewer’s comment/question 2-4.2)

Besides, there is a hysteresis of the magnetization of $Tb_{0.3}Dy_{0.7}Fe_2$ as a function of the magnetic field, and a net magnetization is expected at zero magnetic field. Hence, such a spin volume effect is expected at zero magnetic field, and the volume change ratio expected to follow the magnetization curves as a function of magnetic field. However, these expectations do not agree with the experimental results in this work.

(Authors’ response 2-4.2)

Thank you for the comment. Following the reviewer’s comment, we measured the magnetic field H dependence of d_{peak} again, but we did not see clear hysteresis. In thin-film magnetism, the standard interpretation of an $M-H$ curve that is nearly linear with small hysteresis is that the sample has a small amount of hard magnetic material (with hysteresis) scattered as impurities in a soft magnetic matrix phase with no hysteresis. According to this, the observed $M-H$ curve can be interpreted as some impurity magnetic particles showing hysteresis scattered in the soft magnetic matrix phase of $Tb_{0.3}Dy_{0.7}Fe_2$, which exhibits large magneto-volume effect. The existence of this nominal impurity needs to be confirmed in the future from multiple angles, but it is consistent with the present experimental results, where the magneto-volume effect is considered to be dominated by the $Tb_{0.3}Dy_{0.7}Fe_2$ matrix phase with negligibly small hysteresis. To clarify this point, we added a section to the revised Supplementary Information (Lines 56-64, Page 6).

(Reviewer’s comment/question 2-5.1)

5) The authors observed the opposite trends between Pt and W with respect to the charge current. However, if one looks carefully, there was already a strong signal of Cu/ $Tb_{0.3}Dy_{0.7}Fe_2$ as shown in Fig. 2b. As the spin Hall angle of Cu is negligible, no effect is expected.

(Authors' response 2-5.1)

We thank the reviewer for the comment. Following the comment, we discussed the possible origin of the small but finite mechanical signals in Cu/Tb_{0.3}Dy_{0.7}Fe₂ in the revised main text (Lines 125-130, Page 5). First, spin-Hall angle in Cu is not zero in general; spin current signals in Cu are small but finite in many previous studies [e.g. Uchida, K. *et al.*, J. Appl. Phys. **111**, 103903 (2012), An, H. *et al.*, Nat. Commun. **7**, 13069 (2016), and Okano, G. *et al.*, Phys. Rev. Lett. **122**, 217701 (2019)], being comparable to the present result. As a possibility, we should also consider that the Oersted field originating from the a.c. current flowing in the Cu film induces magnetostriction of the Tb_{0.3}Dy_{0.7}Fe₂ film, but this does not explain the sign reversal in W/Tb_{0.3}Dy_{0.7}Fe₂.

(Reviewer's comment/question 2-5.2)

Furthermore, how about the control sample of a single layer Tb_{0.3}Dy_{0.7}Fe₂? This control experimental would be necessary to further support their claim that the spin current was the major cause.

(Authors' response 2-5.2)

We thank the reviewer for the comment. We totally agree with you. In fact, our first thought was also to make a simple Tb_{0.3}Dy_{0.7}Fe₂ film. However, we realized that it is not possible now, since, in electroplating, a metallic seed film is necessary as a plating electrode. It is not possible to fabricate a single layer of Tb_{0.3}Dy_{0.7}Fe₂. Instead, we designed the control experiment (W/Tb_{0.3}Dy_{0.7}Fe₂ and Cu/Tb_{0.3}Dy_{0.7}Fe₂ samples) that allows us to verify the effects of spin currents, using methods frequently used in past spin current experiments. Following the reviewer's comment, we added a sentence to Methods (Lines 184-185, Page 11).

(Reviewer's comment/question 2-6)

6) Why did the authors select the current frequency to be 10 kHz for their experiment? Did the authors also perform the experiment using other frequencies from low frequency close to the DC regime to high frequencies up to GHz?

(Authors' response 2-6)

We thank the reviewer for the questions. We chose $f_{AC} = 10$ kHz to avoid some technical problems of our laser Doppler system. The measurable frequency range of the laser Doppler vibrometer is <

25 MHz, and it is difficult to access the high frequency (\sim GHz) range of the mechanical signal. Next, the measurable frequency range is further limited by technical issues: in a low frequency range below \sim 1 kHz, mechanical noise becomes ≥ 10 pm, which inevitably appears due to resonant vibration of mechanical parts in the vibrometer and masks the mechanical vibration signals inherent in our samples. On the other hand, the maximum available frequency for reliable measurement is around 12.5 kHz, above which the frequency resolution of the Fourier transform of the vibrometer signal is greater than the FWHM of the observed mechanical peak. Due to these constraints, the available frequency range is $1 \text{ kHz} \lesssim f_{\text{AC}} \lesssim 12.5 \text{ kHz}$ (we chose $f_{\text{AC}} = 10 \text{ kHz}$).

Following the reviewer's comment, as shown in Fig. R4, we measured the mechanical displacement at various frequencies within the available range; the peak exhibits a frequency shift in response to the f_{AC} change. To make this point clear, we added a sentence on the f_{AC} dependence to the main text (Lines 61-62, Page 3) and a section to the Supplementary Information (Lines 48-55, Page 5).

Fig. R4 | Frequency f spectra of signed vibration amplitude d for Pt/Tb_{0.3}Dy_{0.7}Fe₂ at different a.c. current frequencies. The value of j_c and H are 50 mA and 1530 Oe, respectively. The value of f_{AC} was set to 4, 5, 6, 7, 8, 9, and 10 kHz at each measurement.

(Reviewer's comment/question 2-7)

7) It is known that the magnetization dynamics of a ferromagnetic materials is in the ~GHz range, and the spin lifetimes in a typical ferromagnet are usually less than 1 ps. Hence, it is expected the injected spin current will be fully relaxed into the ferromagnet in the 10 kHz regime. As shown in previous spin-orbit torque ferromagnetic resonance measurement, the affected magnetization dynamics due to the spin-Hall torque in Pt/FM bilayer systems also happens in the GHz range.

Following this, does the authors expect the volume change by the 10 kHz AC spin current injection?

(Authors' response 2-7)

Thank you for the comment. Following the reviewer's comment, we added a sentence on the point to the main text (Lines 89-91, Page 4). In fact, the present effect is due to the anti-damping spin torque (Slonczewski spin torque), a bit different from a.c. spin injection such as the spin-torque FMR. The spin-torque FMR generates GHz magnetization dynamics using a GHz range current. On the other hand, the anti-damping spin torque [Slonczewski, J. C., J. Magn. Magn. Mater. **159**, L1 (1996)] modulates GHz magnetization dynamics (and fluctuation) by using a "DC" spin current [e.g. Ando, K. *et al.*, Phys. Rev. Lett. **101**, 036601 (2008)]. In the present model, the input current frequency = 10 kHz is much smaller than that of the magnetization dynamics and fluctuation (~ GHz), and the quasi-d.c. spin current injection, or the anti-damping spin torque, modulates the magnetization fluctuation.

(Reviewer's comment/question 2-8)

8) As the authors claimed that the spin transfer torque is $\sim \mathbf{m} \times (\mathbf{m} \times \boldsymbol{\sigma})$, hence, a zero spin torque is expected if all the \mathbf{m} moments of Tb_{0.3}Dy_{0.7}Fe₂ are parallel to $\boldsymbol{\sigma}$, the spin current direction from Pt.

However, in this work, the authors claimed that the maximum effect is observed when \mathbf{m} is parallel to $\boldsymbol{\sigma}$. This is contrary to the common understanding of spin-transfer torque.

More likely, the torque is applied on the fluctuation parts \mathbf{m} moments that are perpendicular to the external magnetic field direction. The explanation of the statement is suggested to be revised for general audience.

(Authors' response 2-8)

We appreciate the reviewer for the comment. Following the reviewer's suggestion, we added sentences

to the main text (Lines 89-91 and Lines 96-97, Page 4) to explain the detailed physical mechanism behind the spin transfer torque when $\mathbf{m} \parallel \boldsymbol{\sigma}$ (The anti-damping spin torque, or Slonczewski spin torque, acts on the \mathbf{m} fluctuation component, which is maximized at $\mathbf{m} \parallel \boldsymbol{\sigma}$ [Maekawa, S. *et al.*, *Spin Current* (2nd edition). (Oxford University Press, Oxford, 2017), page 99]).

(Reviewer's comment/question 2-9)

9) Does the expansion/suppression of the volume happen along the magnetization direction? In Fig. 1c, the expansion was noted to be perpendicular to the magnetization direction, but in Fig. 1b, it seems that the expansion is the same as the magnetization direction.

(Authors' response 2-9)

Thank you for the question. It is perpendicular to the magnetization in the present configuration. The sample is a film fabricated on a substrate, and, due to the constraints in the film plane on the substrate, the volume change of the thin film is mainly the expansion in the thickness direction. The sample is in-plane magnetized and the displacement is perpendicular to the magnetization. Following the reviewer's comment, we added explanation on this point to the main text (Lines 48-49, Page 2) and the caption to Fig. 1c (Lines 145-146, Page 6).

Reply to the reviewer #3

We thank the reviewer for the valuable comments on our manuscripts. We have carefully revised the manuscripts following the comments.

(Reviewer's comment/question 3-1)

1. Following is my main concern: the current is an AC current. In an applied external magnetic field, this generates a mechanical force in the system at the exact same frequency as the excitation. This mechanical force can induce a motion of the sample that the system will pick up, given its super- μm -level sensitivity. The motion can be amplified by a mechanical resonance, giving rise to the resonant vibrations picked up by the detector. Basically, I wonder if it is not the whole setup that vibrates, rather than the $\text{Tb}_{0.3}\text{Dy}_{0.7}\text{Fe}_2$ that changes thickness. The resistivity of Pt, W and Cu being different, the effects will be different for the different metals.

(Authors' response 3-1)

Thank you very much for the comment. Following the suggestion, we carried out control experiments using (Pt, W, and Cu)/Si samples without the $\text{Tb}_{0.3}\text{Dy}_{0.7}\text{Fe}_2$ layer to double check the roles of the Lorentz force. Figure R5a shows the frequency f spectrum of the signed vibration amplitude d in the Pt/Si sample (a black solid curve) and the Pt/ $\text{Tb}_{0.3}\text{Dy}_{0.7}\text{Fe}_2$ sample (a blue solid curve). No peak appears ($d \sim 0$) in the Pt/Si sample in the absence of the $\text{Tb}_{0.3}\text{Dy}_{0.7}\text{Fe}_2$ layer, while a clear d peak at $f = 10 \text{ kHz} (= f_{\text{AC}})$ appears in the Pt/ $\text{Tb}_{0.3}\text{Dy}_{0.7}\text{Fe}_2$ sample. The result rules out the possibility that the Lorentz force is the origin of the observed mechanical signal in the Pt/ $\text{Tb}_{0.3}\text{Dy}_{0.7}\text{Fe}_2$ sample. We also confirmed that the d_{peak} , d at $f = f_{\text{AC}}$, remains almost zero at various values of H in the Si/Pt sample (black plots in Fig. R5b). As shown in Figs. R5c-f, we also checked the absence of the peaks in W/Si and Cu/Si samples. Note that the sign reversal by replacing Pt with W (W/ $\text{Tb}_{0.3}\text{Dy}_{0.7}\text{Fe}_2$) cannot be explained by the Lorentz force, too. (The value of $f_{\text{AC}} = 10 \text{ kHz}$ is much lower than the typical mechanical resonance frequency of the sample $\gtrsim 1 \text{ MHz}$, and the sample is mechanically off-resonance.) To make this point clear, we added a paragraph to the main text (Lines 121-125, Page 5), and some additional experimental results to the revised Supplementary Information (Lines 12-13 and 20-24, Page 2).

Fig. R5 | Comparison between mechanical signal in (Pt, W, and Cu)/Si and (Pt, W, and Cu)/Tb_{0.3}Dy_{0.7}Fe₂. **a, c, e, f** spectra of d for **(a)** the Pt/Tb_{0.3}Dy_{0.7}Fe₂ (a blue solid curve) and Pt/Si (a black solid curve) samples, **(c)** the W/Tb_{0.3}Dy_{0.7}Fe₂ (a green solid curve) and W/Si (a black solid curve) samples, and **(e)** the Cu/Tb_{0.3}Dy_{0.7}Fe₂ (a grey solid curve) and Cu/Si (a black solid curve) samples. The values of j_c , f_{AC} , and H were set to 50 mA, 10 kHz, and 1530 Oe, respectively. **b, d, f**, The H dependence of d_{peak} , d at $f = f_{AC}$, for **(b)** the Pt/Tb_{0.3}Dy_{0.7}Fe₂ (blue plots) and Pt/Si (black plots) samples, **(d)** the W/Tb_{0.3}Dy_{0.7}Fe₂ (green plots) and W/Si (black plots) samples, and **(f)** the Cu/Tb_{0.3}Dy_{0.7}Fe₂ (grey plots) and Cu/Si (black plots) samples.

(Reviewer's comment/question 3-2)

2. The experiment really purports to measure a change in dimension of the sample in the direction perpendicular to the film plane, so it strictly is a measurement of a linear expansion, not a volume expansion.

(Authors' response 3-2)

Thank you for the comment. As the reviewer pointed out, we measured the thickness change of the Tb_{0.3}Dy_{0.7}Fe₂ film. Following the reviewer's suggestion, we rephrased the title of the paper (Lines 1, Page 1) and the sentences on the experimental configuration in the main text (Lines 44 and 48-52, Page 2).

(Reviewer's comment/question 3-3.1)

3. Is there a difference between this and magnetostriction, an effect well known to exist in the presence of a static magnetic field?

(Authors' response 3-3.1)

We thank the reviewer for the question. The difference is that the present effect is induced by a spin current, while magnetostriction is induced by magnetic fields (The magneto-volume effect is one of various magnetostriction effects). To make this point clear, we added statements to the main text (Line 19, Page 1 and Lines 32-34, Page 2).

(Reviewer's comment/question 3-3.2)

Is this not equivalent to magnetostriction under dynamic spin pumping?

(Authors' response 3-3.2)

Thank you for the question. The present spin injection has been interpreted as the “reverse” process of the “dynamic spin pumping” [Ando, K. *et al.*, Phys. Rev. Lett. **101**, 036601 (2008) and Maekawa, S. *et al.*, *Spin Current* (2nd edition). (Oxford University Press, Oxford, 2017)]. In response to the reviewer's comment, we added a sentence to the Method (Lines 182-184, Page 11).

(Reviewer's comment/question 3-4)

4. Is the absence of an effect when the Pt strip is replaced by a Cu one sufficient to conclude that the effect is not due to magnetostriction in the current's Oersted field? An alternative explanation for the data would then be that, with a Pt film, most of the current flows in the Tb(0.3)Dy(0.7)Fe₂ whereas with the Cu film, most of the current flows in the Cu. The Oersted fields perceived by the Tb(0.3)Dy(0.7)Fe₂ will then be different and therefore the magnetostriction of the Tb(0.3)Dy(0.7)Fe₂ will also be.

(Authors' response 3-4)

Thank you for the comment. The difference Δj_c of the a.c.-current magnitude j_c between the present Pt and Cu films is estimated to be $\Delta j_c/j_c \lesssim 0.01\%$; the resistance of the Tb_{0.3}Dy_{0.7}Fe₂ film is much greater than that of the paramagnetic films (please see the Methods section in the previous manuscript). Therefore, the Oersted field cannot explain the suppression of the mechanical signal in the

Cu/Tb_{0.3}Dy_{0.7}Fe₂ sample. Moreover, please note that the Oersted field scenario is not consistent with the observed sign reversal by replacing Pt with W (W/Tb_{0.3}Dy_{0.7}Fe₂). In response to the reviewer's comment, we added a sentence to the main text (Lines 125-128, Page 5).

(Reviewer's comment/question 3-5)

5. Magnetostriction is even in field. Fig 3b gives the signal as a function of the bias field: the signal is zero without a bias field, and an odd function of that bias field. The presence of a background field in the experiment creates an offset in the strain/field relation, and either the spin current or the Oersted field induce a variation in field around that offset point, which then will give a response that is odd in field. Therefore I am not sure I can uniquely attribute the result to spin injection.

(Authors' response 3-5)

Thank you for the comment. As the reviewer pointed out, an Oersted field is the most worrisome source of contamination. To examine the Oersted field effect, we measured signals in the W/Tb_{0.3}Dy_{0.7}Fe₂ sample because W is typical metal whose sign of the spin Hall effect is opposite to that of Pt [Hoffmann, A. *et al.*, IEEE Trans. Magn. **49**, 5172–5193 (2013)] and found that the W/Tb_{0.3}Dy_{0.7}Fe₂ exhibits the sign reversal of the mechanical displacement (Fig. 2b in the main text), which cannot be explained by the Oersted field effect but consistent with the spin-Hall effect scenario. In response to the reviewer's comment, we added a sentence to the revised main text (Lines 125-128, Page 5).

(Reviewer's comment/question 3-6)

6. There is also electrostriction, typically very large only in ferroelectrics, but non-zero even in electrical conductors. Given the sensitivity of this measurements, how do the authors know it is not this that is measured. Pt, W and Cu probably have very different electrostriction coefficients.

(Authors' response 3-6)

We thank the reviewer for the insightful suggestion. Following the reviewer's suggestion, we carried out some additional experiments: same measurement for (Pt, W, and Cu)/Si samples without the Tb_{0.3}Dy_{0.7}Fe₂ layer. The result (see Fig. R6) shows that the peak signal disappears in the absence of the Tb_{0.3}Dy_{0.7}Fe₂ layer, showing that contribution of the electrostriction effect in Pt, W, or Cu is negligibly small. In response to the reviewer's comment, we added a statement to the revised Supplementary Information (Lines 23-24, Page 2).

Fig. R6 | Mechanical signal in (Pt, W, and Cu)/Si and (Pt, W, and Cu)/Tb_{0.3}Dy_{0.7}Fe₂. **a, b, c,** The H dependence of d_{peak} for **(a)** the Pt/Tb_{0.3}Dy_{0.7}Fe₂ (blue plots) and Pt/Si (black plots) samples, **(b)** the W/Tb_{0.3}Dy_{0.7}Fe₂ (green plots) and W/Si (black plots) samples, and **(c)** the Cu/Tb_{0.3}Dy_{0.7}Fe₂ (grey plots) and Cu/Si (black plots) samples. The values of j_c and f_{AC} were set to 50 mA and 10 kHz, respectively.

[Critical changes are listed below.]

- 1) The main text and the Supplementary Information were revised in response to the comments, which are highlighted in red color.
- 2) Supplementary Note 2, 3, and 4 were added to the Supplementary Information.
- 3) The title of the paper was changed.
- 4) Section headings were added to the main text and the order of Methods, Data availability, Code availability, and Reference was rearranged to follow the format of the journal.
- 5) The reference 26 in the previous main text was changed to the reference 21, and accordingly the order of references was rearranged.
- 6) The following references were added to the main text and the Supplementary Information.

The main text:

27. Uchida, K. *et al.* Thermal spin pumping and magnon-phonon-mediated spin-Seebeck effect. *J. Appl. Phys.* **111**, 103903 (2012).
28. An, H., Kageyama, Y., Kanno, Y., Enishi, N. & Ando, K. Spintorque generator engineered by natural oxidation of Cu. *Nat. Commun.* **7**, 13069 (2016).
29. Okano, G., Matsuo, M., Ohnuma, Y., Maekawa, S., & Nozaki, Y. Nonreciprocal Spin Current Generation in Surface-Oxidized Copper Films. *Phys. Rev. Lett.* **122**, 217701 (2019).

The Supplementary Information:

1. Oomi, G. *et al.*, *J. Magn. Soc. Japan* **23**, 450-452 (1999).
2. Xiao, J., Bauer, G. E. W., Uchida, K., Saitoh, E. & Maekawa, S. Theory of magnon-driven spin Seebeck effect. *Phys. Rev. B* **81**, 214418 (2010).
3. Ishibashi, Y. & Iwata, M. A Theory of Morphotropic Phase Boundary in Solid-Solution Systems of Perovskite-Type Oxide Ferroelectrics. *Jpn. J. Appl. Phys.*, **38**, 800-804 (1999).
4. Shim, H. *et al.* Magnetostrictive performance of electrodeposited $Tb_xDy_{1-x}Fe_y$ thin film with microcantilever structure. *Micromachines* **11**, 523 (2020).
5. Clark, A. E. in *Ferromagnetic Materials*, edited by E. M. Wohlfarth (North-Holland, Amsterdam, 1980), Vol. 1, p. 531.
6. Clark, A. E. Magnetic and Magnetoelastic Properties of Highly Magnetostrictive Rare Earth-Iron Laves Phase Compounds. *AIP Conf. Proc.* **18**, 1015 (1974).
7. Jiles, D. C. The development of highly magnetostrictive rare earth-iron alloys. *J. Phys. D: Appl. Phys.* **27**, 1 (1994).

8. Ren, W. J. and Zhang, Z. D. Progress in bulk MgCu₂-type rare-earth iron magnetostrictive compounds. *Chin. Phys. B* **22**, 077507 (2013).
9. Seki, T. *et al.* Giant spin Hall effect in perpendicularly spin-polarized FePt/Au devices. *Nature Mater.* **7**, 125–129 (2008).
10. Gopman, D. B., Lau, J. W., Mohanchandra, K. P., Wetzlar, K. & Carman, G. P. Determination of the exchange constant of Tb_{0.3}Dy_{0.7}Fe₂ by broadband ferromagnetic resonance spectroscopy. *Phys. Rev. B* **93**, 064425 (2016).

REVIEWER COMMENTS

Reviewer #1 (Remarks to the Author):

This reviewer (second time reviewing this paper) continues to believe the work contained in this manuscript could be useful and represents potentially new important subject matter. Specifically the topic and the approach used is very interesting as well as the results. However, the authors have not provided sufficient information for someone to repeat the results. The authors directed this reviewer to the paper published by Shim et al in 2020 but there are issues with that paper. For example, the composition appeared to be measured by EDX (a better method is WDS) but the EDX approach has issues with TbDyFe due to the overlap of the peaks with iron preventing a clear picture of what the actual composition is. Also the net magnetization in that paper was reported to be 40% lower than the bulk and thus one would believe that the magnetostrictive measurements would be lower. Thus due to the brevity of the authors response from this reviewers first request along with issues regarding the citations the authors are using this reviewer continues to have concerns about the present paper. An easy way to fix this assuming that the main body is constrained is to provide more details in the supplemental information which could be reviewed for this paper.

Reviewer #2 (Remarks to the Author):

The authors have carefully examined my comments, and performed additional experiment to address these issues. Based on the new data and analysis, I think the manuscript has been improved a lot.

Although the conclusion is not 100% supported by the data, I believe that this observation is a new effect that has not been explored yet. And this work will likely intrigue a new research direction to investigate the interplay between spin current and volume. Hence, it is recommended for the publication in Nature Communications.

Reviewer #3 (Remarks to the Author):

After reviewing the response of the authors to my and the other reviewers' comments, I am satisfied that the paper can be published as it is now. The null experiments in response to comments 3-1 about a possible mechanical resonance are satisfying. Ditto for electro-striction, comment 3-6. The revised text addresses comments 3-3 and 3-2 (similar to comment 2-2). The authors had considered the possible effects of an Oersted field (comments 3-4 and 3-5) and eliminated that possibility. The comment about heating (2-3.2) is addressed. The request for a control experiment by reviewer 2 (comment 2-5.2) is also answered. I must add that one can always come up with new possible interpretations for the observations, which then have to be eliminated, but this process has no bounds. At this point, I estimate that the authors have done all that needs to be done before publishing. Future experimentation will confirm or infirm these data and their interpretation, as is normal in the experimental sciences.

Reply to the Reviewer #1

We thank the reviewer for the valuable comments on our manuscripts. Following the comments, we carefully revised the manuscripts as shown below.

(Reviewer's comment/question 1-1)

However the authors have not provided sufficient information for someone to repeat the result.

(Authors' response 1-1)

We thank the reviewer for the comment. Following the reviewer's suggestion, we added explanation on the sample fabrication process to the revised Supplementary Information (Lines 66-74, Page 7). As shown in Fig. R1, we used an electroplating method to fabricate the $\text{Tb}_{0.3}\text{Dy}_{0.7}\text{Fe}_2$ film on the paramagnetic metals. In the method, we prepared solution containing Tb, Dy, and Fe ions and placed a (Pt, W, or Cu)/Si substrate and a Pt wire in the solution as a working electrode and a counter electrode, respectively. We also prepared a reference electrode, enabling us to monitor the voltage between the working and counter electrodes. By adjusting the voltage and deposition time, where the typical deposition rate is ~ 100 nm/hour, we fabricated the $\text{Tb}_{0.3}\text{Dy}_{0.7}\text{Fe}_2$ film with the thickness ~ 100 nm on the paramagnetic metals.

Fig. R1 | Schematic illustration of electroplating deposition of $\text{Tb}_{0.3}\text{Dy}_{0.7}\text{Fe}_2$ film.

(Reviewer's comment/question 1-2)

The authors directed this reviewer to the paper published by Shim et al in 2020 but there are issues with that paper. For example, the composition appeared to be measured by EDX (a better method is WDS) but the EDX approach has issues with TbDyFe due to the overlap of the peaks with iron preventing a clear picture of what the actual composition is.

(Authors' response 1-2)

We thank the reviewer for the comment. Following the reviewer's comment, we remeasured the composition of the present sample by X-ray photoelectron spectroscopy (XPS). From the XPS peaks for Tb, Dy, and Fe at different binding energy, as shown in Fig. R2a, we estimated the atomic composition ratio of the magnetic layer on Pt as Tb:Dy:Fe \sim 11.1:22.5:66.4. From the value, we obtained the film composition \sim Tb_{0.33}Dy_{0.68}Fe_{2.0}, which is close to the optimal composition for large magnetostriction, Tb_{0.3}Dy_{0.7}Fe₂. By measuring the XPS spectra for the magnetic layers fabricated on the W and Cu films (see Figs. R2b and R2c, respectively), we also confirmed the similar composition \sim W/Tb_{0.32}Dy_{0.74}Fe_{1.9} and \sim Cu/Tb_{0.29}Dy_{0.73}Fe_{2.0}. In the revised version, we added sentences on the sample composition and Fig. R2 to the Supplementary Information (Lines 75-81, Page 7).

Fig. R2 | XPS spectra for the magnetic layer fabricated by electroplating deposition. a, b, c XPS peak spectra for the magnetic layer on the Pt film (a), W film (b), and Cu film (c).

(Reviewer's comment/question 1-3)

Also the net magnetization in that paper was reported to be 40% lower than the bulk and thus one would believe that the magnetostrictive measurements would be lower.

(Authors' response 1-3)

Thank you for the comment. As shown in Fig. R3a, the $\text{Tb}_{0.3}\text{Dy}_{0.7}\text{Fe}_2$ film used in the present study was found to exhibit large magnetostriction, which is comparable to the bulk value, in spite of the saturation magnetization smaller than the bulk value, which might be attributable to a finite size effect (Fig. R3b). To avoid confusion, following the reviewer's suggestion, we added explanation on this point and Fig. R3 to the revised Supplementary Information (Lines 81-84, Page 7).

Fig. R3 | Magnetic field H dependence of magnetostriction coefficient and magnetization M of $\text{Tb}_{0.3}\text{Dy}_{0.7}\text{Fe}_2$ film [Shim, H. *et al.*, *Micromachines* 11, 523 (2020)].

(Reviewer's comment/question 1-4)

Thus due to the brevity of the authors response from this reviewers first request along with issues regarding the citations the authors are using this reviewer continues to have concerns about the present paper. An easy way to fix this assuming that the main body is constrained is to provide more details in the supplemental information which could be reviewed for this paper.

(Authors' response 1-4)

We thank the reviewer for the comment. Following the reviewer's suggestions, we added a section on the fabrication process, composition, magnetization, and magnetostriction of the present sample to the revised Supplementary Information (Lines 65-89, Pages 7-8).

Reply to the Reviewer #2

(Reviewer's comment/question 2-1)

The authors have carefully examined my comments, and performed additional experiment to address these issues. Based on the new data and analysis, I think the manuscript has been improved a lot. Although the conclusion is not 100% supported by the data, I believe that this observation is a new effect that has not been explored yet. And this work will likely intrigue a new research direction to investigate the interplay between spin current and volume. Hence, it is recommended for the publication in Nature Communications.

(Authors' response 2-1)

We deeply appreciate reviewer's comments and recommendation for publication in Nature Communications.

Reply to the Reviewer #3

(Reviewer's comment/question 3-1)

The null experiments in response to comments 3-1 about a possible mechanical resonance are satisfying. Ditto for electrostriction, comment 3-6. The revised text addresses comments 3-3 and 3-2 (similar to comment 2-2). The authors had considered the possible effects of an Oersted field (comments 3-4 and 3-5) and eliminated that possibility. The comment about heating (2-3.2) is addressed. The request for a control experiment by reviewer 2 (comment 2-5.2) is also answered. I must add that one can always come up with new possible interpretations for the observations, which then have to be eliminated, but this process has no bounds. At this point, I estimate that the authors have done all that needs to be done before publishing. Future experimentation will confirm or infirm these data and their interpretation, as is normal in the experimental sciences.

(Authors' response 3-1)

We deeply appreciate reviewer's comments and recommendation for publication in Nature Communications.

[Critical changes are listed below.]

- 1) The main text and the Supplementary Information were revised in response to the comments, which are highlighted in red color.
- 2) Supplementary Note 5 was added to the Supplementary Information.
- 3) The references 4 and 5 in the previous Supplementary Information were changed to the references 2 and 3, respectively, and accordingly the order of references was rearranged.

REVIEWERS' COMMENTS

Reviewer #1 (Remarks to the Author):

The authors have responded to all of my queries and this reviewer appreciates the authors time on this matter. This reviewer has one additional minor request. Can the authors add into the supplemental section on how the magnetostriction was measured. This is a non-trivial measurement on thin film and the test approach should be described.

Reply to the Reviewer #1

We thank the reviewer for the valuable comments on our manuscripts. Following the comments, we carefully revised the manuscripts as shown below.

(Reviewer's comment/question 1-1)

This reviewer has one additional minor request. Can the authors add into the supplemental section on how the magnetostriction was measured. This is a non-trivial measurement on thin film and the test approach should be described.

(Authors' response 1-1)

We thank the reviewer for the comment. Following the reviewer's suggestion, we added a section on the magnetostriction measurement of the $\text{Tb}_{0.3}\text{Dy}_{0.7}\text{Fe}_2$ film to the revised Supplementary Information (Lines 111-120, Page 10). To evaluate magnetostriction of the $\text{Tb}_{0.3}\text{Dy}_{0.7}\text{Fe}_2$ film, as shown in Fig. R1, we fabricated a tri-layer $\text{Tb}_{0.3}\text{Dy}_{0.7}\text{Fe}_2/\text{Cu}/\text{Si}$ cantilever and measured the mechanical displacement of the cantilever in the magnetic field H [Shim, H. *et al.*, *Micromachines* **11**, 523 (2020)]. By analyzing the mechanical displacement along the y -axis in response to H along the z -axis, we obtained the H dependence of the magnetostriction coefficient of the $\text{Tb}_{0.3}\text{Dy}_{0.7}\text{Fe}_2$ film.

Fig. R1 | Schematic illustration of magnetostriction measurement of $\text{Tb}_{0.3}\text{Dy}_{0.7}\text{Fe}_2$ film.